# The Interactions Between HBV and the Innate Immunity of Hepatocytes

**DOI:** 10.3390/v12030285

**Published:** 2020-03-05

**Authors:** Fayed Attia Koutb Megahed, Xiaoling Zhou, Pingnan Sun

**Affiliations:** 1Stem Cell Research Center, Research Center for Reproductive Medicine, Guangdong Provincial Key Laboratory of Infectious Diseases and Molecular Immunopathology, Shantou University Medical College, Shantou 515041, China; fayed.attia@yahoo.com; 2Department of Nucleic Acid Researches, Genetic Engineering and Biotechnology Research Institute, General Autority-City of Scientific Researches and Technological Applications, Alexandria 21934, Egypt

**Keywords:** HBV, innate immunity, hepatocytes, in-vitro cell models

## Abstract

Hepatitis B virus (HBV) infection affects ~350 million people and poses a major public health problem worldwide. HBV is a major cause of cirrhosis and hepatocellular carcinoma. Fewer than 5% of HBV-infected adults (but up to 90% of HBV-infected infants and children) develop chronic HBV infection as indicated by continued, detectable expression of hepatitis B surface antigen (HBsAg) for at least 6 months after the initial infection. Increasing evidence indicates that HBV interacts with innate immunity signaling pathways of hepatocytes to suppress innate immunity. However, it is still not clear how HBV avoids monitoring by the innate immunity of hepatocytes and whether the innate immunity of hepatocytes can be effective against HBV if re-triggered. Moreover, a deep understanding of virus–host interactions is important in developing new therapeutic strategies for the treatment of HBV infection. In this review, we summarize the current knowledge regarding how HBV represses innate immune recognition, as well as recent progress with respect to in vitro models for studying HBV infection and innate immunity.

## 1. Introduction

Chronic infection with hepatitis B virus (HBV) is the main cause of liver cirrhosis and hepatocellular carcinoma (HCC) worldwide. Although there is a prophylactic vaccine to prevent HBV infection, HBV still infects ~350 million people and contributes toward viral hepatitis-associated morbidity and mortality. The HBV genome consists of a partially double-stranded DNA molecule approximately 3.2 kb long that replicates via an RNA intermediate [1,2,3]. Basically, the innate immune system responds to viral infection in three phases. Firstly, various sensors in the cytoplasm recognize pathogen-associated molecular patterns, such as foreign DNA or RNA, and send a warning message to initiate proinflammatory and antimicrobial responses by activating a multitude of intracellular signaling pathways, including adaptor molecules, kinases, and transcription factors. The second phase involves the proteins of the downstream signaling pathways transmitting the danger message to the nucleus to activate effector elements. Finally, the consequently up-regulated effectors (i.e., inflammatory factors or interferon (IFN)-stimulated genes) degrade exogenous viral elements such as viral DNA, RNA, and proteins [4].

It is widely accepted that the adaptive immune responses play vital roles in the clearance of HBV infection. Early innate immune response is essential for the following induction of adaptive immunity. Restoring or boosting innate immunity linked to adaptive immunity may help eradicate chronic HBV infection. However, the role of innate immunity during HBV infection appears not to be well understood, which can be attributed to the fact that the recruitment of patients in the very early, asymptomatic phase of HBV infection is very difficult [5,6]. HBV belongs to the hepadnavirus family and acts as a “stealth” virus because it does not induce obvious innate immune responses such as type I and II interferons (IFNs) in the early stage of infection [7]. Moreover, many in vitro cell and animal models have proved that the target cells (hepatocytes) do not recognize HBV efficiently through known innate immune signaling in the acute infection of HBV, indicating the possibility of an HBV immune evasion mechanism [8]. In addition, HBV also has the ability to suppress functions of innate immune cells [9,10].

In this review article, we summarize the recent knowledge and debate regarding HBV and innate immunity signaling pathways in hepatocytes, the different HBV proteins that target innate immunity, and different advanced in vitro cell models for the study of virus–host interactions. Collation of such information will provide new insights into novel therapeutic treatments for chronic hepatitis B infection (CHB).

## 2. Genetic Organization, Life Cycle, and Global Epidemiology of HBV

HBV is a member of the hepadnavirus family. The genetic organization HBV is shown in Figure 1. It has a relaxed partially double-stranded circular DNA genome of about 3200 bases with four main overlapping open reading frames (ORFs) encoding surface protein (pre-S/S), pre-core/core (pre-C/C), transcriptional co-activator (X), and DNA polymerase (P) genes [11]. The preS/S ORF encodes three different, enveloped structural proteins termed large (L), middle (M), and small (S) proteins. The S protein (HBsAg) consists of 226 amino acids (aa), and the M protein has an extra N-terminal extension of 55 aa, whereas the L protein has a further N-terminal extension of 108 or 119 aa depending on the HBV genotype. The preC/C ORF codes for two distinct products: the core protein forming the nucleocapsid protein shell (hepatitis B core antigen (HBcAg)) and the precore protein that derives from in-frame alternative initiation sites. The X ORF encodes the small regulatory X protein, which is essential for viral replication. The P ORF encodes protein P, the viral DNA polymerase [12]. HBV relies on protein P, which is also a specialized reverse transcriptase (RT), to replicate its genomic DNA via an RNA intermediate [13]. Protein P consists of four domains: a terminal protein that is covalently linked to the DNA primer during negative-strand DNA synthesis, a spacer domain that is tolerant to mutations, the RT domain, and the ribonuclease H (RNase H) domain [14].

After entry into hepatocytes, the HBV genome is delivered into the nucleus where it is repaired to form covalently closed circular DNA (cccDNA) that serves as a template for the pregenomic RNA (pgRNA) and subgenomic RNAs for translation to HBV proteins (including the viral polymerase protein (Pol), HBsAg, hepatitis B X protein (HBx), the core protein, and the precore protein [15]). The precore protein contains the entire sequence of the core protein plus an amino-terminal extension of 29 amino acids (i.e., the “precure” sequence) [16]. The first 19 amino acids of the precore protein constitute a signal peptide that directs the precore protein to the endoplasmic reticulum (ER) for secretion. This signal peptide is removed by the signal peptidase, located in the ER lumen, to generate the 22-kDa precore protein derivative p22, which is further cleaved at its carboxy terminus by furin protease in the Golgi. The secreted pre-core protein derivative is known as the e antigen called HBeAg. After the formation of the viral core particle, the HBV pgRNA is reverse-transcribed by the viral DNA polymerase that is also packaged to become the circular and partially double-stranded DNA genome. Finally, the core particles subsequently interact with HBsAgs in intracellular membranes for the formation of mature viral particles, which are then released from infected hepatocytes to infect other cells as shown in Figure 2 [17].

Globally, chronic hepatitis B infection affects around 350 million people worldwide (3.61% of the global population), resulting in 600,000 deaths annually from cirrhosis, liver failure, and hepatocellular carcinoma [18]. The number of affected individuals is the highest in the Western Pacific region (defined by the World Health Organization as the 37 countries including China, Japan, South Korea, the Philippines, and Vietnam), with 95.3 million infected (prevalence estimates of 5.26%), and Africa, with 75.6 million infected (prevalence estimates of 8.83%), accounting for nearly 70% of all chronic hepatitis B infection cases globally [19]. The percentages of liver cirrhosis and HCC caused by HBV infection are 30% and 45% worldwide, respectively [20], while the percentages in China are 60% and 80% [21].

## 3. HBV Infection and Host Innate Immunity

Innate immunity is the first line of defense against microbial pathogens, including viruses. Viral infection triggers the induction of type-I interferons (e.g., IFN-α and IFN-β) and other proinflammatory cytokines through two distinct signaling pathways [22].

One of these pathways utilizes a subfamily of Toll-like receptors (TLR3, 7, 8, and 9) to detect HBV nucleic acids in the endosome after the endocytosis of viral particles. These TLRs are localized in the endosomal membranes of specialized cell types, such as plasmacytoid dendritic cells (pDCs) [23], and they recruit the adaptor protein MyD88 or Toll-Like receptor adaptor molecule 1 (TRIF) to activate protein kinases, including the inhibitor of nuclear factor kappa B kinase (IKK) complex (consisting of IKKα, IKKβ, and NEMO/IKKγ) and the IKK-related kinases (TANK Binding Kinase 1 (TBK1) and inhibitor of nuclear factor kappa B kinase subunit epsilon (IKKε)). The IKK complex phosphorylates the nuclear factor kappa B subunit 1 (NF-κB) inhibitor (IκB) and targets IκB for degradation by the ubiquitin proteasome pathway, thereby allowing NF-κB to enter the nucleus to induce a large array of genes involved in immune and inflammatory responses [24]. TBK1 and IKKε phosphorylate IFN-regulatory factors, IRF3 or IRF7, resulting in its dimerization and nuclear translocation [25]. The nuclear IRFs, NF-κB, and other transcription factors form an enhanceosome complex to activate the expression of interferons [26], which are then secreted to bind to their receptors on HBV-infected as well as neighboring non-infected cells. The engagement of interferon receptors activates the Janus kinase (JAK)-signal transducer and activator of transcription (STAT) signaling pathway to induce interferon-stimulated genes (ISGs), which suppress HBV replication and assembly (Figure 3) [27].

The other HBV/innate immunity signaling pathway utilizes the retinoic acid inducible gene I (RIG-I) to detect HBV double-stranded RNA (dsRNA) in the cytosol [28]. RIG-I binds to HBV dsRNA through its C-terminal RNA helicase domain and mediates the activation of IKK and TBK1/IKKε through its N-terminal caspase activation and recruitment domains (CARD). The adaptor protein that links RIG-I to IKK and TBK1/IKKε activation is the recently identified mitochondrial antiviral signaling protein (MAVS), also known as IPS-1, VISA, or CARDIF. MAVS contains an N-terminal CARD domain that interacts with the tandem CARD domains of RIG-I and a C-terminal transmembrane domain that localizes it to the mitochondrial outer membrane [29]. Cell culture studies have suggested that MAVS/IPS-1 is required for interferon induction by cytosolic DNA [30].

The production of pro-inflammatory cytokines and IFNs and the activation of natural killer (NK) cells is frequently observed during the early phase of viral infections. Previously, HBV was considered as a “stealth virus” that could establish persistent infection in hepatocytes by evading the host innate immune system [31]. Using an experimentally infected chimpanzee model, HBV was unable to interfere with host cellular gene transcription and induce ISG expression in the liver [7]. However, by quantification of serum cytokines, a study which enrolled 21 HBV-infected patients during the pre-symptomatic phase indicated that HBV infection did not elicit strong production of IFNs and interleukin (IL)-15, but did induce the production of the anti-inflammatory cytokine IL-10 [32]. In addition to this observation, another study suggested that many cytokines are weakly induced during acute HBV infection. After initiation of viral expansion and before the peak of viremia, IFN-α, tumor necrosis factor (TNF)-α, IL-15, IL-10, IL-6, and IL-1β levels were detectable in serum samples from about half of HBV patients [33].

On the other hand, in another study Mutz et al. elucidated the interactions between HBV and innate immunity to better understand the mechanisms of immune activation in HBV infection and reported that HBV does not trigger the IFN response of hepatocytes or interfere with the innate immune-sensing functions of hepatocytes, and nor does it inhibit the IFN-stimulated pathways of hepatocytes based on absence of interferon and cytokine production [34]. On the other hand, macrophages can be activated by high-titer HBV to mount an immune response against HBV mainly through production of inflammatory cytokines like TNF-α and IL6 [35]. In contrast, several other studies have reported that HBV replication can inhibit those aforementioned pattern-recognition receptor (PRR) functions. One study showed that HBeAg-positive chronic hepatitis B patients had downregulated TLR2 expression in the hepatocyte plasma membrane [36]. Other studies based on heterologous overexpression of viral proteins and/or host sensors have also addressed this issue, but have yielded conflicting results [37,38]. Similarly, contradictory data exist regarding whether HBV infection suppresses the cellular response to IFN. One study demonstrated the inhibition of signal transducer and activator of transcription 1 (STAT1) nuclear import by HBV polymerase, and showed that HBsAg and/or HBx protein interfered with the STAT1 signaling [39].

In conclusion, these studies indicate that HBV can target the TLR system and thus attenuate the anti-HBV responses of the innate immune system, which implies that HBV does not affect at least some aspects of the innate immunity system. HBV, particularly HBx protein, was reported to disrupt RIG-I-mediated IFN-β induction by downregulating MAVS [40,41]. HBx was recently reported to act as a deubiquitinating enzyme which deubiquitinated RIG-I and other molecules including TNF receptor associated factor 3 (TRAF3), IRF3, and IKKi. It attenuated the interaction between RIG-I and TRAF3 that plays an important role in IFN induction, and finally dampened type I IFN induction [42]. In addition, HBV proteins also interfere with JAK-STAT signaling and ISG expression. For example, HBV polymerase was shown to inhibit nuclear translocation of STAT1 [43] and HBV precore/core proteins inhibited myxovirus resistance A (MxA) gene expression via their interaction with the MxA promoter [44]. All these findings provide evidence that HBV can counteract the innate immune responses mediated by TLRs or RIG-I in the liver microenvironment, which might be strategies by which HBV escapes the surveillance of the host innate immune system. The different HBV protein contributions in induction of cellular innate immunity are summarized in Table 1.

### 3.1. Hepatitis B e Antigen (HBeAg)

TLR expression on Kupffer cells, peripheral monocytes, and hepatocytes is reduced in HBeAg-positive CHB compared with patients with HBeAg-negative CHB [36]. However, HBV cDNA plasmids encoding the G1896A precore stop codon mutation (which abrogates production of HBeAg) have no effect on TLR2 expression [51]. More recently it has been shown that HBeAg directly interacts with key adapters in the TLR2 pathway [52]. The expression of IFN-α and IFN-β mRNA was down-regulated in stably transformed HBeAg-positive HepG2 cells as compared to an HBeAg-negative HepG2 cell line [53]. Thus, it can be concluded that HBeAg protein downregulates TLR innate immunity signaling pathway.

### 3.2. HBV Core Protein (HBcAG) and Hepatitis B Splice Protein (HBSP)

HBV core protein interfered with IFN-β expression through binding to the IFN-β promoter as a transacting silencer in murine fibroblasts [54]. Subsequently, Rosmorduc et al. showed that HBV core protein, encoded by spliced pgRNA, down-regulated IFN-inducible MxA protein expression, with MxA being an important antiviral protein kinase [55]. Further to these findings, Soussan et al. showed that in addition to the core and precore protein, the most frequently detected spliced pgRNA also encodes a novel protein termed HBSP that may down-regulate MxA [56]. Thus, it can be concluded that HBcAg and HBSP down-regulates IFN-inducible MxA protein expression.

### 3.3. Hepatitis B Virus X Protein (HBx)

Previous reports have also shown that HBx promotes MAVS ubiquitination to trigger its proteasome-mediated degradation through the Lys136 site, and that MAVS K136R elicits a higher level of IFN-β activation compared with wild type MAVS, suggesting that the MAVS Lys136 site could be the ubiquitination site targeted by the ubiquitin E3 ligase RNF125 [40], enabling MAVS proteins to undergo proteasomal degradation by RNF125-mediated ubiquitin conjugation [57]. PSMA7 is a subunit of the proteasome that regulates the activity of this complex associated with HBX, suggesting that HBX may modulate the function of the proteasome by interacting with PSMA7. PSMA7 may regulate host innate immune signaling by destabilizing MAVS, raising the possibility that HBX may be potentially bridged by PSMA7 on the mitochondrial outer membrane to exert its inhibitory effect on innate immune response [40]. In addition, another research group has proposed a new model where the HBx protein blocks tripartite motif containing 22 (TRIM22) transcription, leading to a decrease in interferon (IFN)-induced TRIM22 expression [48].

### 3.4. Hepatitis B Virus Polymerase Inhibits Innate Immunity (HBV Pol)

Three observations are worthy of mention to discuss the impact of HBV Pol on IFN signaling. First, previous data indicated the inhibitory effect of HBV Pol on IRF signaling and to a lesser extent on NF-κB signaling, which also contributes to IFN-β production. Second, abundant detection of HBV Pol in the nonencapsidated state indicates that HBV Pol could contribute to viral pathogenesis or immune evasion. Lastly, HBV Pol has been previously identified as one of the viral proteins that confers resistance to IFN treatment [45]. Previous studies have shown that HBV polymerase is a potent inhibitor of IFN-β induction in human hepatocytes, and its expression leads to inhibition of promoter activity and transcription of IFN-β and antiviral immunity in the PH5CH8 primary hepatocytic cell line. In addition, it has been demonstrated that HBV polymerase interferes with IFN-β induction at the TBK1/IKKε level, expression of HBV polymerase inhibits Sendai virus (SeV)-induced endogenous IRF3 phosphorylation, dimerization, and nuclear translocation, and RNA helicases of the DEAD-box protein family (DDX3) may be involved in the inhibition of IFN-β induction by HBV polymerase [46].

## 4. Conflicts Regarding HBV and Innate Immunity

In the current review, we set out to elucidate the interactions between HBV and innate immunity to better understand the mechanisms of immune activation in HBV infection. Previously, HBV was considered a “stealth” virus that did not interfere with the innate immune-sensing functions of hepatocytes. However, data obtained from recent studies suggest that circulating innate immune cells, as well as liver cell populations, can sense and respond to HBV infection, which enables the innate immune system to detect and restrict the invading virus. In addition, it is worth noting that HBV induced IFN responses in hepatocytes are relatively weak as compared with other viral infections, which is consistent with the observations from the studies obtained in chimpanzee [7] and mouse models [35]. The previous reports highlight that HBV is recognized by host PRRs and thus induces innate immune responses that decrease HBV replication and expansion. However, the specific PRRs and intracellular signaling pathways involved in HBV recognition and inhibition still require further investigation. An in-depth understanding of immune mechanisms induced by distinct components of HBV will provide the opportunity to characterize the immunopathogenesis of HBV infection and develop immune based therapeutic strategies for HBV infection [58,59].

In contrast, several other studies have reported that HBV replication can inhibit those aforementioned PRR functions. One study showed that HBeAg-positive chronic hepatitis B patients have downregulated TLR2 expression on hepatocytes [60]. One study demonstrated inhibition of STAT1 nuclear import by HBV polymerase [39] and another reported that the HBsAg and/or HBx protein interfered with the STAT1 signaling [52]. By stimulating ex vivo-culture liver biopsies with different TLRs ligands and SeV, it was observed that induction of the innate immune response, as measured by IFN and ISG expression, did not differ between HBV-infected and non-infected samples, suggesting that HBV infection neither induces nor interferes with the innate immune response, which is consistent with the observed lack of innate immune response during acute HBV infection in experimentally infected chimpanzees and the apparent sensitivity of HBV to TLR-mediated induction of innate immunity as such, in contrast to many other viruses including hepatitis C virus (HCV) and hepatitis A virus (HAV). These results support the hypothesis that HBV behaves like a “stealth virus” by staying under the radar of the pathogen detection system [61].

We believe that contradictory reports are mainly related to three aspects. The first is the infection stage. It is important to find evidence of innate immune responses in the early phase of acute hepatitis B infection even though no innate immune responses are observed in patients with chronic HBV. The second aspect is related to the experimental models. It is critical how similar the innate immunity pathways are in these models and primary human hepatocytes (PHH). The third aspect relates to the HBV genotypes. There are differences in innate immunity response among different HBV genotypes.

## 5. In Vitro Models to Study HBV Infection and Innate Immunity

To date, HBV research has been hampered by a distinct lack of robust infectious model systems that both support productive HBV infection and accurately mimic virus–host interactions. Hepatoma-derived cell lines support HBV replication and particle assembly following the transfection of cloned viral genomes [62,63]. After differentiation, differentiated hepatic cells derived from a human hepatic progenitor cell line (HepaRG) become susceptible to HBV, indicating expression of the HBV receptor in the differentiated HepaRG and the differentiation-dependent expression of an HBV receptor. It took a long time until Wenhui Li’s group from Beijing discovered the human sodium taurocholate cotransporting polypeptide receptor (NTCP) to be the long sought-after HBV receptor [64]. This finding was confirmed and consistent with the pronounced induction of NTCP that is unique to the hepatocyte membrane and expression upon differentiation of HepaRG cells [65]. Prior studies have shown that primary human hepatocytes support HBV infection, although infection is usually not robust even upon supplementation of cell-culture medium with dimethyl sulfoxide or polyethylene glycol [65]. Moreover, primary human hepatocytes rapidly lose their hepatic phenotype shortly after isolation from the in vivo microenvironment [66]. Several drawbacks have been found in these in vitro HBV infection models. In the case of HepaRG cells, for example, it is impossible to evaluate the effects of genetic background on HBV infection using HepaRG cells. There is a restricted availability of primary human hepatocytes (PHHs), although PHHs isolated from young children can proliferate in humanized chimera mice [66]. Besides the efforts in generating HBV-susceptible cell lines, independent approaches have involved the generation of hepatocyte like cells (HLCs) through the differentiation of stem cells from diverse origins, such as human embryonic stem cells (hESCs), human induced-pluripotent stem cells (hiPSCs), liver-resident hepatic progenitor cells, or bone marrow-derived mesenchymal stem cells [67,68,69,70].

However, there is thus an urgent need for a novel in vitro HBV infection model. Recently, human hepatocyte-like cells differentiated from hESCs and hiPSCs have gained much attention not only due to their promise for regenerative medicine, but also due to their potential for modeling drug metabolism and pathogen infection in vitro [71,72,73,74]. For example, human hepatocyte-like cells differentiated from these stem cells can be stably supplied due to the indefinite proliferation potential of hESCs and hiPSCs. Influences on the genetic background of the cells can be evaluated because iPS cells are established from various types of somatic cells. In addition, genome editing technology makes gene knockout and gene replacement possible in hESCs and hiPSCs [75]. The protocols established for the production of HLCs from hESCs usually consist of three crucial steps: (1) endodermal induction, (2) hepatic specification, and (3) hepatocyte maturation. So far, a gain in susceptibility to HBV has not been a criterion for the functionality of HLCs. In a former study, Shlomai and co-workers showed for the first time that HBV can infect hiPSC-derived HLCs. They demonstrated that expression of NTCP becomes strongly induced during hepatic maturation, which correlates with a gain in susceptibility to HBV. This finding demonstrates the potential of human hiPSC-derived HLCs for in vitro studies of HBV biology. Importantly it opens the door for generating HBV-susceptible cells from individual groups of patients and individuals with certain genetic polymorphisms [76]. We thus consider that iPS-HCLs are promising as an alternative in vitro model of infection by hepatotropic pathogens, including HBV. Different in-vitro HBV cell models, their HBV infection efficiencies, innate immunity pathways, advantages, and limitations are summarized in Table 2.

## 6. Conclusions

The innate immune system is the first line of host defense against HBV infection and plays an important role in anti-HBV. HBV only triggers little innate response in host cells after infection and has evolved escape strategies from the innate immune system. HBV proteins, including HBx, HBV polymerase, HBs, HBc, and HBe, block TLRs, the Jak-Stat pathway (type I/III IFN response), TBK1/IKKε (the effector kinase of the IRF-3/NF-κB pathway), and cytokines.

Current evidence obtained from experiments has highlighted the importance of innate immunity in the early control of HBV spread. However, how HBV succeeds to avoid innate immune recognition has still not been fully clarified. The recent development of varied novel models, such as HepaRG cells, stem-cell derived hepatocytes, and liver organoids, may greatly contribute to clarifying the interaction between HBV and host innate immunity. Given the central role that innate immunity plays in antiviral responses, the enhancement of HBV-induced innate immunity may provide fundamental insights into the therapy of HBV. We believe that a better understanding of HBV–host interactions will be achieved by using these novel models, eventually leading to effective treatment strategies for hepatitis B.

## Figures and Tables

**Figure 1 viruses-12-00285-f001:**
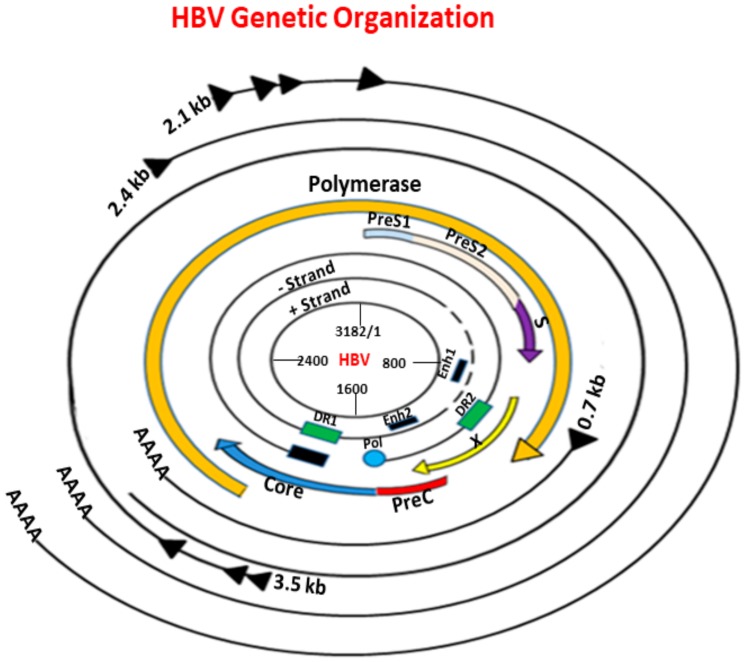
Genetic organization of the hepatitis B virus (HBV). HBV contains a small, partially double-stranded DNA (dsDNA) of about 3.2 kb. The main four open reading frames (ORFs) are shown: precore/core (preC/C) that encodes the e antigen (HBeAg) and core protein (HBcAg), P for polymerase (reverse transcriptase), Pre-surface/surface (preS/S) including preS1/preS2/S for surface proteins (small (S), middle (M), and large (L)), and X for transcriptional trans-activator protein. The genome contains four promoters, two enhancer regions (Enh1, Enh2), and two direct repeats (DR1, DR2). The outer lines represent the different classes of HBV mRNA transcripts.

**Figure 2 viruses-12-00285-f002:**
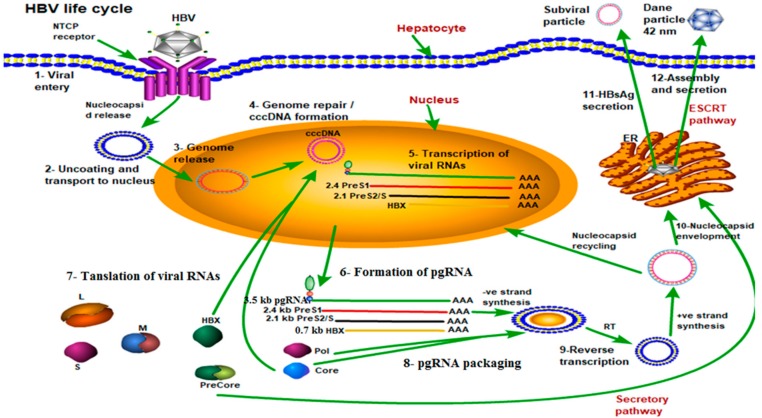
Life cycle of the HBV shown by green arrows step by step. Mature HBV virions enter hepatocytes through the sodium taurocholate cotransporting polypeptide receptor (NTCP) on the cell membrane (Step 1). After release from the viral envelope, the nucleocapsid is then transported to the nucleus where the genome is repaired to form covalently closed circular DNA (cccDNA) (Steps 2, 3 and 4). Using cccDNA as the template, viral RNAs are transcribed and exported into the cytoplasm where they are translated to form the viral proteins (Steps 5, 6 and 7). Additionally, pregenomic RNA (pgRNA) is packaged by core protein, along with the polymerase protein, and the viral genome is replicated through reverse transcription (RT) of the pgRNA to form the - strand, followed by partial synthesis of the + strand (Steps 8 and 9). Mature nucleocapsids can then either be recycled back to the nucleus to maintain a pool of cccDNA, or enveloped and secreted through the endosomal sorting complexes required for transport (ESCRT) pathway (Steps 10, 11 and 12).

**Figure 3 viruses-12-00285-f003:**
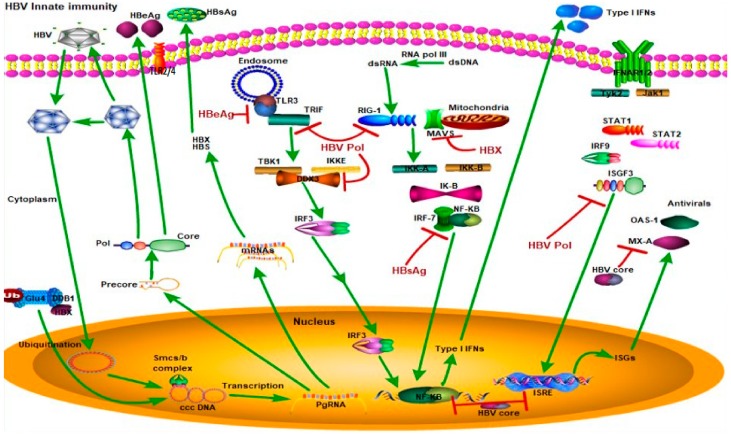
HBV suppression of type I interferon (IFN) response (Green arrows indicate the related signaling pathways and T-arrows in red indicate blocking the targets). Different HBV proteins block the type I IFN response. For example, hepatitis B surface antigen (HBsAg) inhibits interferon regulatory transcription factor 7 (IFR7) and nuclear factor kappa B subunit 1 (NF-κB) interaction to block NF-κB translocation to the nucleus and block type 1 IFN production. Hepatitis B e antigen (HBeAg) blocks toll-like receptor 3 (TLR3) binding to toll-like receptor adaptor molecule 1 (TRIF) to suppress the TLR3 pathway. HBV core protein inhibits production of type I IFNs and interferon-stimulated response elements (ISREs). Hepatitis B X protein (HBX) proteins inhibit binding of mitochondrial antiviral signaling protein (MAVS) to mitochondria to block the RIG1-MAVS pathway. HBV polymerase inhibits binding of DEAD-Box Helicase 3 X-Linked (DDX3) to the TBK1/IKKE complex, inhibits the RIG-I pathway, and blocks interferon-stimulated gene factor 3 (ISGF3) to inhibit ISRE production. Abbreviations—TLR3: Toll-like receptor 3, TRIF: Toll-like receptor adaptor molecule 1, TBK1: TANK-binding kinase 1, IKKE: inhibitor of nuclear factor-kB kinase, IRF3: interferon regulatory transcription factor 3, NF-κB: nuclear factor kappa B, dsRNA: double-stranded RNA, dsDNA: double-stranded DNA, MAVS: mitochondrial antiviral signaling protein, RIG-I: retinoic acid-inducible gene I, IRF-7: interferon regulatory transcription factor 7, IFNAR1/2: interferon alpha receptor 1/2, TYK2: tyrosine kinase 2, JAK1: Janus kinase 1, STAT1/2: signal transducer and activator of transcription 1/2, IRF-9: interferon regulatory transcription factor 9, ISGF3: interferon-stimulated gene factor 3, ISRE: interferon-stimulated response element, ISGs: interferon-stimulated genes, OAS1: 2’-5’ oligoadenylate synthetase 1, MX-A: myxovirus resistance A.

**Table 1 viruses-12-00285-t001:** Different cellular innate immunity targets of HBV infection.

HBV Proteins	Cellular Innate Immunity Targets	References
Polymerase	RIG-I, TLR3/TBK1, IKKɛ, DDX3/IFN-β, IFN/JAK-STAT	[45,46,47]
Hepatitis B virus X protein	RIG-I, melanoma differentiation-associated gene 5 (MDA5)/MAVS/IFN-β/Trim22	[40,41,48,49]
Core/precore	IFN/myxovirus resistance A (MxA)	[50]
Hepatitis B e antigen	TLR2/ MyD88-Adapter-Like (MAL)	[51,52]

**Table 2 viruses-12-00285-t002:** Innate immunity pathways in HBV cell models.

Cell Models	HBV Infection Efficiency	Innate Immunity Pathways Present	Advantages Limitations	References
Primary human hepatocytes (PHHs)	Average 50% in the presence of 5% polyethylene glycol (PEG)	Low TLR expression	Gold standard for investigation of HBV infection; limited availability and unpredictable variability	[77,78,79]
		Low stimulator of interferon genes(STING) expression		
		RIG-I/MDA5		
		NF-κB pathway		
		IRF pathway		
		IFN pathway		
Human embryonic stem cell (hESC)/human induced-pluripotent stem cell (hiPSC)-derived hepatocytes	25–90%	Low TLR expression	Close to PHHs depending on differentiation status; can support long-term infection, can be generated from donors with different genetic backgrounds; immature status needs to be improved	[72,76,80,81,82]
		Low STING expression		
		NF-κB pathway		
		IRF pathway		
		IFN pathway		
HepaRG	~10% in the presence of PEG, maximum rate is 20%	Low TLR expression	Close to PHHs;	[78,79,83,84]
		Low STING expression	suitable for drug metabolism and HBV infection; can be differentiated into both biliary cells and hepatocytes; a long period for differentiation is needed (at least two weeks)	
		RIG-I/MDA5		
		NF-κB pathway		
		IRF pathway		
		IFN pathway		
HepG2-NTCP	~70% infection efficiency at 4% PEG and 2.5% dimethyl sulfoxide (DMSO)	Poorly characterized	Can be used to screen novel drugs and elucidate host–virus interaction; high concentrations of HBV genome equivalents are needed for a high infection rate	[37,64,65,79,85,86,87,88,89,90]
		TLR expression		
		No cyclic GMP-AMP synthase (cGAS)		
		expression and low STING expression		
		RIG-I/MDA5		
		NF-κB pathway		
		IRF pathway		
		IFN pathway		
Huh7-NTCP	~5% infection efficiency at 4% PEG and 2.5% DMSO	Poorly characterized	Low HBV infection rate; defects in some innate immunity pathways	[37,64,65,79,88,89,90]
		TLR expression		
		No cGAS and STING expression		
		Both RIG-I/MDA-5 and IFN pathway are present but weaker than HepG2		
HepG2.2.15	HBV genome integrated the into HepG2 genome	No cGAS expression and low	Can be used to screen anti-HBV drugs Unsuitable for studying HBV entry and uncoating	[37,62,86,87,91,92]
		STING expression		
		RIG-I/MDA5		
		NF-κB pathway		
		IRF pathway		
		IFN pathway		

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
