# Peer review of "The Interactions Between HBV and the Innate Immunity of Hepatocytes"

_viruses, 2020, doi:10.3390/v12030285_

Round 1

Reviewer 1 Report

Dr. Megahed and colleagues presented an highly interesting review article in which they discussed important aspects of Interactions between HBV and the innate imunity.

The authors describe current knowledge about the interplay between HBV and the host’s innate immune system during infection and have pointed out the conflict of ideas or opinions on this subject matter. The colleagues have also highlighted current in-vitro models for the study of HBV.

Overall, the review article is mainly comprehensive, appropriate referenced, and concise in its content. The introduction section introduces the following sections well. The review article is informative and interesting to read. The figures and tables are well constructed, concise, and easy to follow. The main and important aspect of the current knowledge of hepatitis B virus infection and innate immunity has been well addressed. In general this is an attractive review article. However, there are minor comments which should be addressed.

Comments:

  • A continuous flow of the ideas is somehow missing. I suggest to rewrite so that the information contained therein can be more easily assessed by the reader,
  • There are a number of abbreviations which has not been stated in full. Therefore, the abbreviation should be stated first and abbreviation used subsequently, except for conventionally recognised acronyms.

Author Response

We have made revision in line42 ‘And early innate immune response is essential for the following induction of adaptive immunity. Restoring or boosting innate immunity linked to adaptive immunity may help eradicate chronic HBV infection.’ and line 277 ‘We think the controversial reports may mainly derive from three aspects. The first one is the infection stage. It is important to find the evidence of innate immune responses in the early phase of acutely infected hepatitis B infection even though no innate immune responses observed in patients with chronic HBV. The second one is the experimental models. It is critical how similar the innate immunity pathways in these models and PHH. The third one is the HBV genotypes. Actually, there is difference in innate immunity response among different HBV genotypes.’

We have full name of the abbreviation when it firstly appears.  

Reviewer 2 Report

It is an interesting Review about the interactions between HBV and Innate Immunity of Hepato-cytes. Innate immunity is important associated HBV infection, also innate immunity linked to adaptive immunity is important is important. Therefore, authors should refer to HBV infection associated with adaptive immunity.

 Chronic hepatitis B (CHB) show a defective early innate immune response, which are essential for the further induction of HBV-specific adaptive immunity and may contribute to the persistence of CHB or a weakened capacity to clear HBV. Viral loads were previously shown to affect the quality of the anti-HBV immune responses and outcomes of viral infections. The levels of hepatitis and viral replication observed in various phases of CHB are considered to reflect distinct states of virus-host immune interactions. Early innate evasion by HBV may be important for the initial establishment of infection, rather than the persistence of HBV infection. The actual elimination of HBV infection requires the presence of adaptive immune responses. HBV may initially evade innate immune recognition to establish infection. On the other hand, upon exposure to high-level HBV, human macrophage could be activated with increased inflammatory cytokine expression. Changes in the balance of cytokine profiles may result in either persistence of HBV infections. CHB with HBeAg (+) with high viral loads is more strongly associated with the activation of Th1-and Th2-type responses than CHB with HBeAg (-). Thus, preferential activation and commitment towards Th1- or Th2-cell subsets may influence the clinical consequences of HBV infection. The mechanism by which HBV evades innate immune recognition and establishes persistent infection remains a subject of debate. Besides, some researchers are becoming more interested in how to eradicate chronic HBV infection by restoring or boosting innate immunity linked to adaptive immunity.

Author Response

According to the reviewer’s comments, we added in our manuscript with ‘And early innate immune response is essential for the following induction of adaptive immunity. Restoring or boosting innate immunity linked to adaptive immunity may help eradicate chronic HBV infection.’ in line 42.

Reviewer 3 Report

This is good review while some modification is required before publishing.

Major concerns:

1, Figure 2 needs to be clearer, the author failed to describe the process of mRNA translation while added “7 translocation of viral proteins”. This looks like these proteins are not derived from virus mRNA. Also, not all proteins are translocated into the nucleus, so the title translocation of viral proteins should not be put upon all proteins.

2, Figure 3 only show TLRs on endosome membrane. However, TLR2 and TLR4 are also enriched cell membrane and HBeAg can interact with TLRs there.

Minor concern.

“IκB kinase complex” in Line 123 should be “IKK complex”.

Author Response

Response: In Figure 2, we have revised ‘7. Translocation of viral proteins’ into ‘7. Translation of viral RNAs’, ‘pgRNA’ into ‘3.5 kb pgRNA’, ‘HBx’ into ‘0.7 kb HBx’.

Response: We have added TLR2/4 in the hepatocyte membrane and HBeAg can interact with them there in Figure 3.

Response: We have revised ‘IkB kinase complex’ in Line 123 into ‘IKK complex’ according to the reviewer’s comment.

Reviewer 4 Report

Manuscript ID: viruses-734004

Type of manuscript: Review

Title: The Interactions between HBV and Innate Immunity of Hepatocytes

Authors: Fayed Attia Koutb Megahed et al. ,

This is an interest review.

Which do authors support, the presence or the absence that HBV interacts innate immunity? Did the controversial reports depend the experimental models?

Author Response

Good questions.

In our opinion, we think HBV does interact innate immunity of hepatocytes. According to our data in HepG2-NTCP, HepG2.2.15 and stem-cell derived hepatocytes (unpublished), HBV replication level was regulated by modulating the innate immunity of these cells. The innate immunity of hepatocytes restrict HBV replication. On the other hand, HBV infection triggered a slight innate immunity response in hepatocytes.   

First, we think the controversial reports depend the experimental models. Largely depending on the how similar the innate immunity pathways in these models and PHH.

Second, the controversial reports also depend the infection stage. We think HBV interact innate immunity of hepatocytes at the initial infection. And in the chronic infection stage, HBV may suppress innate immunity. No difference in IFN and ISG expression of liver specimens between chronic HBV patients and control (refer to: Suslov, A.; Boldanova, T.; Wang, X.; Wieland, S.; Heim, M. H., Hepatitis B Virus Does Not Interfere With Innate Immune Responses in the Human Liver. Gastroenterology 2018, 154, 1778-1790.) Furthermore, it is important to find the evidence of innate immune responses in the early phase of acutely infected hepatitis B infection.

Third, we think the controversial reports depend on the HBV genotypes. Some genotypes may trigger higher innate immunity response than others (Refer to: Sato, S.; Li, K.; Kameyama, T.; Hayashi, T.; Ishida, Y.; Murakami, S.; Watanabe, T.; Iijima, S.; Sakurai, Y.; Watashi, K.; Tsutsumi, S.; Sato, Y.; Akita, H.; Wakita, T.; Rice, C. M.; Harashima, H.; Kohara, M.; Tanaka, Y.; Takaoka, A., The RNA Sensor RIG-I Dually Functions as an Innate Sensor and Direct Antiviral Factor for Hepatitis B Virus. Immunity 2015, 42, 123-132.).